# Synthesis of Blue-Emissive InP/GaP/ZnS Quantum Dots via Controlling the Reaction Kinetics of Shell Growth and Length of Capping Ligands

**DOI:** 10.3390/nano10112171

**Published:** 2020-10-30

**Authors:** Woosuk Lee, Changmin Lee, Boram Kim, Yonghyeok Choi, Heeyeop Chae

**Affiliations:** 1School of Chemical Engineering, Sungkyunkwan University (SKKU), 2066, Seoburo, Suwon 16419, Korea; ws.lee@skku.edu (W.L.); cm2nara@naver.com (C.L.); bobohn5310@skku.edu (B.K.); Yong684@skku.edu (Y.C.); 2SKKU Advanced Institute of Nanotechnology, Sungkyunkwan University (SKKU), 2066, Seoburo, Suwon 16419, Korea

**Keywords:** non-Cd blue emitters, InP/Ga/ZnS quantum dots, fatty acid, heating-up synthesis

## Abstract

The development of blue-emissive InP quantum dots (QDs) still lags behind that of the red and green QDs because of the difficulty in controlling the reactivity of the small InP core. In this study, the reaction kinetics of the ZnS shell was controlled by varying the length of the hydrocarbon chain in alkanethiols for the synthesis of the small InP core. The reactive alkanethiol with a short hydrocarbon chain forms the ZnS shell rapidly and prevents the growth of the InP core, thus reducing the emission wavelength. In addition, the length of the hydrocarbon chain in the fatty acid was varied to reduce the nucleation kinetics of the core. The fatty acid with a long hydrocarbon chain exhibited a long emission wavelength as a result of the rapid nucleation and growth, due to the insufficient In–P–Zn complex by the steric effect. Blue-emissive InP/GaP/ZnS QDs were synthesized with hexanethiol and lauryl acid, exhibiting a photoluminescence (PL) peak of 485 nm with a full width at half-maximum of 52 nm and a photoluminescence quantum yield of 45%. The all-solution processed quantum dot light-emitting diodes were fabricated by employing the aforementioned blue-emissive QDs as an emitting layer, and the resulting device exhibited a peak luminance of 1045 cd/m^2^, a current efficiency of 3.6 cd/A, and an external quantum efficiency of 1.0%.

## 1. Introduction

Colloidal quantum dots (QDs) have received significant attention as promising emitting materials for next-generation display devices owing to their outstanding color purity, convenient color tunability, near-unity photoluminescence quantum yield (PLQY), and low-cost solution processability [1,2,3,4,5,6]. Although cadmium (Cd) chalcogenide QDs have been extensively studied over the past few decades [7,8,9,10,11,12], their intrinsic toxicity has strictly limited any commercial application [13]. Indium phosphide (InP) QDs are considered to be an environmentally friendly alternative because of their relatively low toxicity and emission wavelength covering most visible wavelengths. The synthesis of InP QDs is, however, relatively challenging compared with that of Cd QDs because of their covalent character, vulnerability to oxidizing environments, and lattice mismatch between the InP core and ZnS shell. The optical properties of InP QDs have been improved by the introduction of inner shells such as ZnSe [14,15,16], ZnSeS [17,18,19,20], and GaP [21,22,23] to alleviate abrupt interfacial strain and the removal of the oxide surface of the InP core through in situ etching [15,16]. However, the development of blue-emissive InP QDs still lags behind that of red and green QDs because of the difficulty in controlling the reactivity of the small InP cores. The small InP cores are also difficult to passivate because of their large surface-to-volume ratio [24,25,26].

One of the popular approaches employed to obtain blue-emissive InP QDs is the introduction of Ga with a relatively wide band gap (GaP, 2.27 eV) into the InP core. The In^3+^ to Ga^3+^ cation exchange reaction was used to adjust the composition of the InGaP core, and the emission wavelength was controlled by the degree of Ga alloying into the InP core [21,27,28,29,30]. A small InP core was obtained by etching with hydrogen fluoride or acetic acid; however, the optical properties were poor due to the easy oxidation of the naked core [31]. The reaction kinetics of the small core have also been controlled by using relatively less reactive phosphorus (P) precursors, such as tris(triphenylsilyl)phosphine and tris(dimethylamino)phosphine instead of tris(trimethylsilyl)phosphine, and through competitive nucleation of the InP core and Cu_3−*x*_P in the presence of copper ions [32,33,34]. However, most of these studies were conducted using the hot-injection method rather than the simple and reproducible heating-up synthesis. The hot-injection method requires rapid and homogeneous mixing of reagents at high temperatures to synthesize monodisperse nanocrystals. As the volume of the reaction solution increases, it is difficult to maintain a uniform temperature and concentration of the reagents in the reactor during counter ion injection, thus hindering reproducibility. In contrast, the heating-up method has advantages in terms of scaling-up and reproducibility because the reaction proceeds after uniform mixing of all reagents [35].

In this study, the reaction kinetics of the ZnS shell was controlled by varying the hydrocarbon chain length of the alkanethiols to synthesize blue-emissive QDs. The relatively more reactive alkanethiols (the sulfur source) are expected to facilitate the formation of ZnS shells and retard InP core growth. The hydrocarbon chain length of alkanethiols was varied from 6 to 12, and the emission wavelength of the resulting QDs was investigated. The hydrocarbon chain length of the fatty acids (the capping ligands) was additionally controlled to reduce the reaction kinetics to facilitate the synthesis of small InP cores. The hydrocarbon chain length of the fatty acids was varied from 12 to 18, and the temporal evolution of the UV–visible (UV–vis) absorption and photoluminescence (PL) spectra of the aliquots was compared. A blue-emissive QD was obtained by adopting hexanethiol as the sulfur source and lauric acid as the capping ligand, and their structural and optical properties were characterized. Thus, all-solution processed quantum light-emitting diodes (QLEDs) were successfully fabricated by employing the above blue-emitting QDs, and their performance was investigated.

## 2. Materials and Methods

### 2.1. Reagents

All chemicals, indium acetate (In(OAc)_3_, 99.99% trace metal basis, Sigma-Aldrich, St. Louis, MO, USA), zinc acetate (Zn(OAc)_2_, 99.99%, Sigma-Aldrich), lauric acid (98%, Sigma-Aldrich), myristic acid (99%, Sigma-Aldrich), stearic acid (99%, Sigma-Aldrich), oleic acid (90% technical grade Sigma-Aldrich), gallium trichloride (GaCl_3_ beads, anhydrous 99.999% trace metals basis, Sigma-Aldrich), 1-hexanethiol (95%, Sigma-Aldrich), 1-octanethiol (≥98.5%, Sigma-Aldrich), 1-dodecanethiol (≥98%, Sigma-Aldrich), 1-octadecene (ODE, 90% technical grade, Sigma-Aldrich), and tris(trimethylsilyl)phosphine ((TMS)_3_P, 10 wt% in hexane, SK Chemicals, Gyeonggi-do, Korea), were used without any further purification.

### 2.2. Synthesis of InP/GaP/ZnS QDs

InP/GaP/ZnS QDs containing fatty acids with different carbon chain lengths were synthesized according to a previously reported method with some modifications [23]. The schematic diagram of the synthesis is shown in Figure 1. First, 0.48 mmol of In(OAc)_3_, 1.99 mmol of Zn(OAc)_2_, and 4.34 mmol of the fatty acid (i.e., lauryl acid (LA), myristic acid (MA), palmitic acid (PA), or stearic acid (SA)) were mixed in 8 mL of ODE and degassed at 120 °C under a vacuum for 2 h to prepare the In precursor solution. The solution was cooled to room temperature after placing it in a nitrogen atmosphere. Subsequently, 0.38 mmol of (TMS)_3_P (the source of P) and 0.17 mmol of GaCl_3_ beads were mixed in 2 mL of ODE in a glove box and sufficiently dissolved. After adding the above (TMS)_3_P and GaCl_3_ solutions to the In precursor solution and stirring for 10 min, the mixture was rapidly heated to 300 °C. The faster the heating rate, the more uniformly the precursor is converted to monomer over a shorter period of time, contributing to a narrower emission wavelength. Therefore, all the syntheses were performed by preheating the mantle for 15 min, which corresponds to a heating rate of 26.1 °C∙min^−1^. The alkanethiol (i.e., hexanethiol (HT), octanethiol (OT), or dodecanethiol (DDT)), which is the sulfur source, was injected into the reaction mixture quickly at an elevated temperature (185–275 °C). Zinc oleate (6 mmol) and DDT (3.0 mL) were also added when the temperature reached 300 °C for further shell growth, and the reaction was continued for 10 min. The crude solution was precipitated with ethanol and centrifuged to remove excess free ligands, unreacted precursors, and impurities. Finally, the precipitate was re-dispersed in a hexane–octane mixture for further analysis, such as UV–vis, PL, absolute PLQY, transmission electron microscopy (TEM), and device fabrication.

### 2.3. Synthesis of Blue-Emissive InP/GaP/ZnS QDs

The blue-emissive InP/GaP/ZnS QD was synthesized using the aforementioned scheme after parametrization. Here, LA and HT were selected as the capping ligands and sulfur source, respectively, and HT was injected at 185 °C. After the addition of zinc oleate (6 mmol) and DDT (3.0 mL) at 300 °C, the reaction was continued for 60 min instead of 10 min. The QD solution in the hexane–octane mixture was obtained through the same purification process.

### 2.4. Fabrication of QLEDs

Conventional QLEDs with a structure of indium tin oxide (ITO) cathode/PEDOT:PSS/PVK:poly-TPD/QDs/ZnO/Al were fabricated to evaluate the device characteristics. A patterned ITO glass substrate with a sheet resistance of approximately 10–15 Ω∙sq^−1^ was cleaned sequentially with deionized water, acetone, and methanol with a bath sonication for 15 min and was further treated with oxygen plasma for 10 min. The PEDOT:PSS (Clevios AI 4083, Leverkusen, Germany) hole injection layer was spin-coated at 3000 rpm for 60 s and annealed at 150 °C for 15 min in air; it was then moved to a glove box filled with nitrogen gas (O_2_ < 1 ppm, H_2_O < 1 ppm). A mixture of PVK (dissolved in chlorobenzene at a concentration of 10 mg/mL) and poly-TPD (dissolved in chlorobenzene at a concentration of 10 mg/mL) was deposited on the PEDOT:PSS layer, followed by annealing at 180 °C for 20 min. The sky blue InP/GaP/ZnS QD solution was then spin-coated at 3000 rpm for 60 s, followed by a spin-coating of ZnO (dissolved in ethanol at a concentration of 100 mg/mL) at 1500 rpm for 120 s. Finally, an Al electrode (∼120 nm) was deposited at a deposition rate of 5–8 Å/s by a thermal evaporation process under a high vacuum of ∼6 × 10^−6^ torr. The fabricated devices were evaluated immediately without encapsulation.

### 2.5. Characterization

UV–vis absorption spectra and PL emission spectra of the QD solution were measured using a UV–visible spectrophotometer (JASCO, V630, Tokyo, Japan) and a fluorescence spectrometer (Agilent Technologies, G9803AA, Santa Clara, CA, USA) at room temperature. The absolute PLQY of the QDs was measured using an absolute PLQY spectrometer (QD-2100 from Otsuka Electronics Co., Ltd., Osaka, Japan). Inductively coupled plasma-optical emission spectrometry (ICP-OES) (Agilent Technologies, 5100, Santa Clara, CA, USA) and X-ray diffraction (XRD) measurements (Bruker, D8 ADVANCE, Karlsruhe, Germany) were conducted to analyze the structure of the QDs. High-resolution transmission electron microscopy (HR-TEM) (JEOL, JEM-ARM 200F, Tokyo, Japan) images of the QDs were obtained at an accelerating voltage of 200 kV. The current density–voltage–luminance (*J*–*V*–*L*) characteristics of the QLEDs were measured using a spectroradiometer (Konica-Minolta, CS-2000, Tokyo, Japan) coupled with a voltage–current source measurement unit (Tektronix, Keithley 2400, Beaverton, OR, USA). Device evaluations were carried out without encapsulation in air under ambient atmosphere.

## 3. Results and Discussion

### 3.1. The Emission Wavelength Depending on the Sulfur Source Type and Injection Temprerature

The emission spectra of the QDs prepared with different alkanethiols were compared to investigate the effect of hydrocarbon chain length on the reaction kinetics of ZnS shell growth, as presented in Figure 2a. HT, with the shortest carbon chain, exhibited the shortest PL emission wavelength of 497 nm with a full width at half-maximum (FWHM) of 59 nm in comparison with 510 nm emitted by QDs prepared using OT and DDT. These results indicate that a short and reactive alkanethiol forms the ZnS shell rapidly and prevents the growth of the InP core, thus reducing the emission wavelength. S-TOP (element sulfur dissolved in trioctylphosphine) is commonly used for ZnS shell formation; the PL emission peak of the QD prepared with S-TOP shifted to a longer wavelength of 517 nm, indicating that S-TOP slows down the ZnS shell growth due to lower reactivity, resulting in further growth of the core. The asymmetric PL spectrum observed in the case of S-TOP is believed to originate from incomplete surface passivation [36,37]. Since the time point of formation of the ZnS shell can play a role in determining the size of the InP core, the effect of the injection temperature of alkanethiols on the optical properties of the QDs was also investigated, as shown in Figure 2b. The PL emission peak increased gradually from 491 nm at the injection temperature of 185 °C to 545 nm at 275 °C, and the FWHM widened significantly from 54 nm to 68 nm with increasing injection temperature. These results can be attributed to the core growth through Ostwald ripening until the shell was formed, leading to a large size distribution. Therefore, short-length alkanethiol with a low injection temperature results in the rapid formation of the ZnS shell, resulting in shorter wavelength QDs.

### 3.2. The Effect of the Capping Ligand Length on the Kinetics of Nucleation

Next, the effect of the capping ligand length on the kinetics of nucleation and growth of InP QDs was investigated by monitoring the temporal evolution of the UV–vis absorption and PL spectra of the aliquots, as shown in Figure 3. At 100 °C, the first excitonic absorption peak was observed for SA, whereas no absorption peak was observed for LA, MA, and PA ligands, indicating that nucleation does not occur at or below 100 °C (Figure 3a,b). It was also observed that the shorter the hydrocarbon chain length of the capping ligands, the shorter the first excitonic transition peak, indicating a smaller core size. The PL emission spectra of the above aliquots are shown in Figure 3c,d. The QDs with SA ligands exhibited a relatively long PL peak wavelength of 478 nm at 180 °C, which increased to 521 nm at 300 °C owing to rapid nucleation and growth. In contrast, the QDs with LA ligands exhibited the shortest PL peak wavelength of 466 nm at 180 °C. These results are in good agreement with the absorption spectrum results. However, these findings contradicted results from a previous study, which indicated that the longer the hydrocarbon chain, the slower the nucleation and growth reaction [38].

It is well known that the InP core is formed by In–P ligand complexes at approximately 100 °C, but stable In–P–Zn intermediate complexes are also formed at mild temperatures when Zn precursors are introduced, resulting in a retarded nucleation of the InP core [35,39,40]. We believe that the length of the capping ligand affects the formation of the In–P–Zn complexes, resulting in a difference in the reactivity of the core. For further verification, we compared the temporal evolution of the UV–vis absorption spectra of the QDs synthesized using oleic acid (OA), which has the same hydrocarbon chain length as that of SA, to reconfirm the effect of steric hindrance by long ligands on the reactivity of the core, as shown in Figure 4. The first excitonic absorption peak for the QD with OA at 100 °C (Figure 4a) was comparable to that of the SA ligand (Figure 4b) and increased with a similar trend as the reaction progressed. Therefore, we believe that the steric hindrance caused by longer capping ligands interferes with the formation of the In–P–Zn complex, resulting in rapid nucleation and growth of the core.

### 3.3. Optical and Structural Analyses of Blue-Emissive QDs

Blue-emissive InP/GaP/ZnS QDs were synthesized utilizing LA ligands and HT as the sulfur source. The optical properties and structure of the synthesized blue-emissive QDs were characterized, as shown in Figure 5. The resulting QDs exhibited a PL peak at a wavelength of 485 nm with a line width of 52 nm (Figure 5b). However, the QDs showed a low absolute PLQY of 45% due to the insufficient stabilization of the short capping ligand (LA) [41,42]. The synthesized QDs had an average size of approximately 6.3 nm and a spherical shape, as shown in the TEM image (see Figure 5a). No clear boundary between the core and shell was observed in the image, indicating that the GaP inner shell contributed to the epitaxial coating of the ZnS shell. Powder XRD and ICP-OES analyses were performed to confirm the composition of the QDs in each reaction step, as shown in Figure 5c,d. In the XRD pattern, the InP core showed a cubic zinc blende structure and no peak change was observed despite the introduction of the GaP inner shell due to the negligible thickness of the GaP shell. The cubic ZnS peak was clearly identified for the InP/GaP/ZnS QD, indicating that a ZnS shell of an appropriate thickness was well formed on the core. ICP-OES was performed to confirm the presence of the GaP inner shell. Ga was identified in the InP/GaP intermediate and InP/GaP/ZnS QDs, indicating that the GaP inner shell sequentially grew on the tiny InP core before the formation of the ZnS shell.

### 3.4. Applications of Blue-Emissive QDs in QLEDs

The QLEDs with a conventional device architecture were fabricated by all-solution processing, as shown in Figure 6a and their performance was investigated. The blended hole transport layer (HTL) of poly-TPD and PVK was employed owing to the advantages of the fast mobility of poly-TPD and the high energy level of PVK, which is more suitable for InP-based QLEDs, as reported in our previous studies [43]. The device exhibited an electroluminescence (EL) peak emission at 491 nm with a FWHM of 66 nm (at 7 V), which was slightly red-shifted by 6 nm compared with the PL spectra (485 nm) owing to the well-known quantum confinement Stark effect (Figure 6b) [44]. No parasitic spectra from neighboring HTLs (i.e., P-TPD and PVK) were observed, except for the emission spectrum of the QDs, indicating that carrier recombination occurred only in the QD layer. The *J*–*V*–*L* characteristics of the QLEDs and the dependence of current efficiency (CE) and external quantum efficiency (EQE) on the current density are illustrated in Figure 6c,d. The resulting device exhibited a low turn-on voltage of 2.5 V and a maximum brightness of 1016 cd/m^2^ at an applied voltage of 7 V (corresponding to a current density of 180 mA/cm^2^). The maximum CE and EQE achieved was 3.6 cd A^−1^ and 1.0%, respectively.

## 4. Conclusions

The reaction kinetics of the ZnS shell were controlled by varying the hydrocarbon chain length of alkanethiols, used as a sulfur source, to obtain a small core for blue-emissive QDs. It was found that short alkanethiols led to rapid formation of the ZnS shell due to its high reactivity, thus resulting in a shorter emission wavelength for the QD. The wavelength of the PL emission peak of the QDs with HT was 497 nm, whereas that of the QDs with OT and DDT was 511 nm. The hydrocarbon chain length of the fatty acids employed as the capping ligands was additionally controlled to examine the reaction kinetics of the InP core; this, to the best of our knowledge, was investigated for the first time in this study. The short hydrocarbon capping ligands formed a stable In–P–Zn ligand complex rather than an In–P complex in the presence of the Zn precursor, resulting in delayed In–P nucleation. The resulting QDs exhibited a PL emission peak wavelength of 510 nm; the QD prepared with a long hydrocarbon capping ligand (SA) displayed the peak at 521 nm. These results are attributed to the steric hindrance of the capping ligands affecting In–P–Zn ligand complex formation in the presence of the Zn precursor. The blue-emissive InP/GaP/ZnS QD was synthesized using LA and HT, and the resulting QDs exhibited a PL peak wavelength of 485 nm with a line width of 52 nm. QLEDs with a conventional device architecture were successfully fabricated by employing the aforementioned blue-emissive QDs. The QLEDs exhibited a peak luminance of 1045 cd/m^2^, a CE of 3.6 cd/A, and an EQE of 1.0%. This study demonstrates that the reaction kinetics can be controlled by varying the hydrocarbon chain length of the alkanethiols and fatty acids to obtain blue-emissive InP QDs by heating-up synthesis.

## Figures and Tables

**Figure 1 nanomaterials-10-02171-f001:**
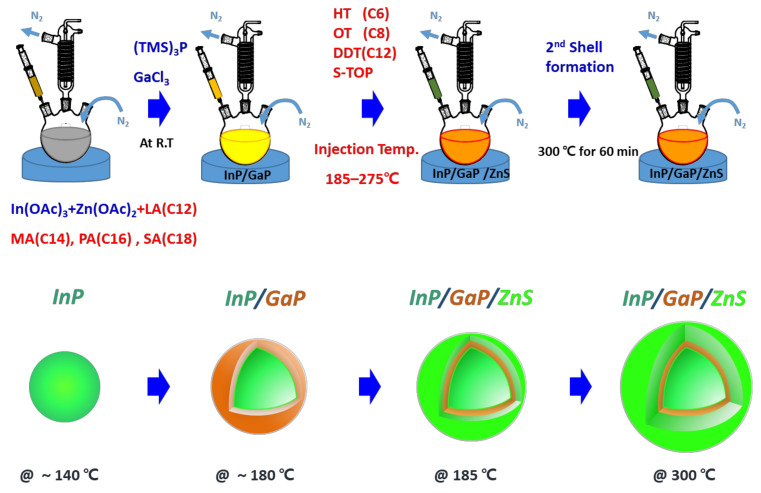
The schematic of the reaction scheme is presented in the top panel. The expected quantum dot (QD) structure at each reaction step is shown in the panel below.

**Figure 2 nanomaterials-10-02171-f002:**
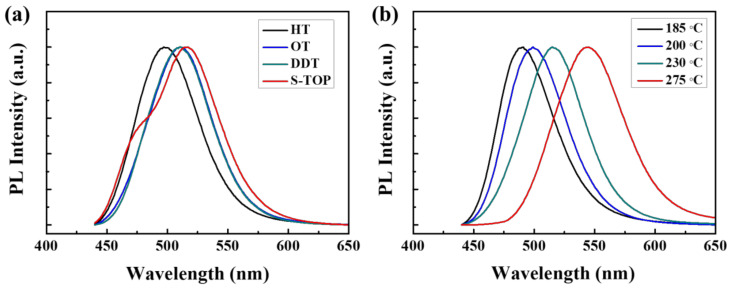
Photoluminescence emission spectra of the QDs prepared (**a**) with different sulfur sources (injected at 230 °C under the same reaction conditions) and (**b**) at different injection temperatures of octanethiol.

**Figure 3 nanomaterials-10-02171-f003:**
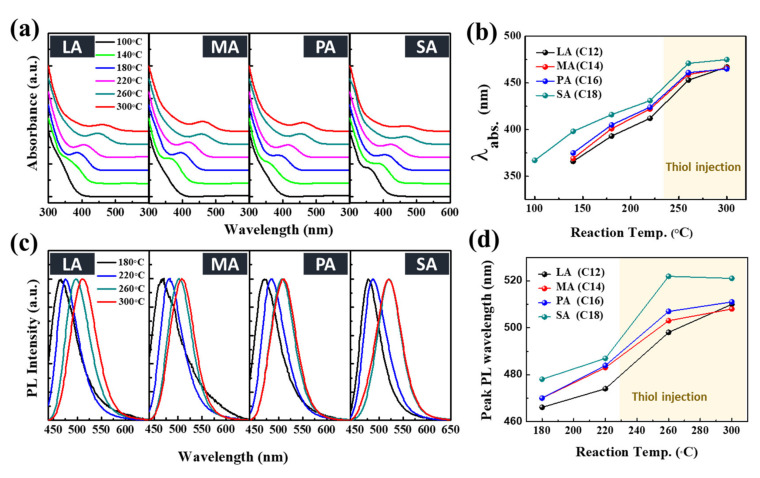
(**a**) Temporal evolution of UV–vis absorption spectra of InP/GaP/ZnS QDs with lauryl acid (LA), myristic acid (MA), palmitic acid (PA), and stearic acid (SA) as the capping ligands and (**b**) their first excitation transition wavelengths, which were determined from second derivatives of the spectra. (**c**) photoluminescence (PL) emission spectra and (**d**) the PL peak wavelength of the corresponding QDs.

**Figure 4 nanomaterials-10-02171-f004:**
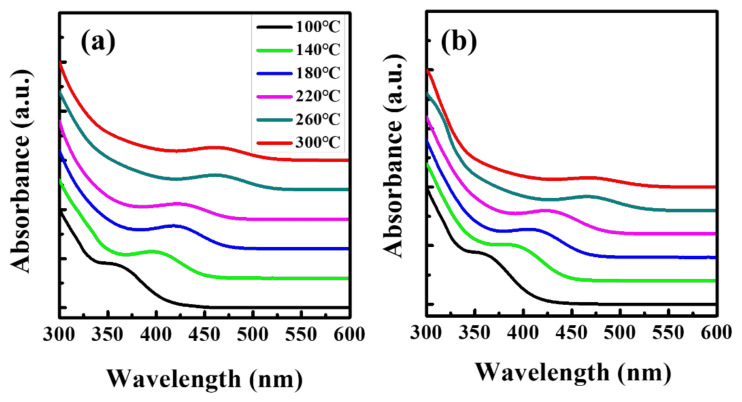
Temporal evolution of the UV–vis absorption spectra of InP/GaP/ZnS QDs prepared with (**a**) oleic acid and (**b**) stearic acid.

**Figure 5 nanomaterials-10-02171-f005:**
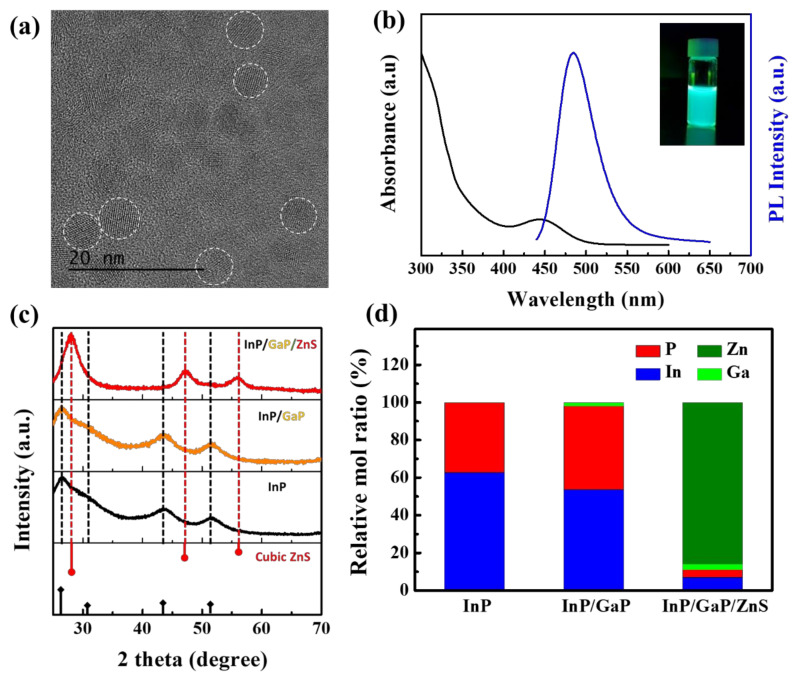
(**a**) TEM images and (**b**) normalized PL and absorption spectrum of the blue-emissive InP/Ga/ZnS QDs. (**c**) XRD pattern (the corresponding bulk reflections of the cubic zinc blende InP and cubic ZnS are shown in the lower panels) and (**d**) relative molar ratios of InP, InP/GaP, and InP/GaP/ZnS QDs analyzed by ICP-OES.

**Figure 6 nanomaterials-10-02171-f006:**
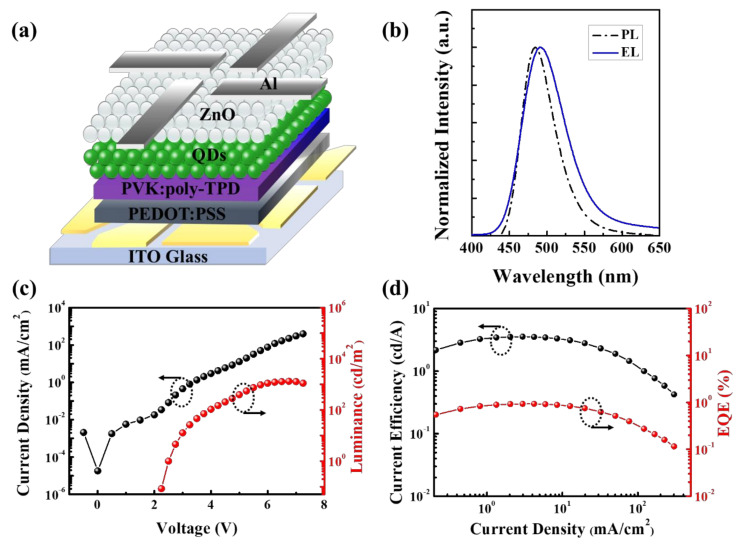
(**a**) Schematic of the multilayered all-solution processed quantum light-emitting diode (QLED), (**b**) spectral comparison of the PL of a QD dispersion with electroluminescence (EL) collected at 7 V, (**c**) current density–voltage–luminance (*J*–*V*–*L*) characteristics, and (**d**) current efficiency (CE) and external quantum efficiency (EQE) as a function of the current density of the QLED.

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
