# Peer review of "Synthesis of Blue-Emissive InP/GaP/ZnS Quantum Dots via Controlling the Reaction Kinetics of Shell Growth and Length of Capping Ligands"

_nanomaterials, 2020, doi:10.3390/nano10112171_

Round 1

Reviewer 1 Report

The paper is devoted to the synthesis and characterization of colloidal InP/ZnS core-sell quantum dots currently considered as a promising candidate for many applications including display and biolabeling.

Authors have suggested and realized controlling ZnS shell growth rate by the variation in capping ligand chain length. As a result, they succeeded in the manufacturing of InP/GaP/ZnS core-sell quantum dots having a tiny InP core and emitting at the wavelength as short as <500 nm.

The paper contains rather new results, is distinctly written and seems to be of interest for colloidal chemistry and semiconductor communities.

The only remark is that the small-size InP core inside the ZnS shell has a lattice mismatch of 7.7% and should be compressed depending on the shell thickness. Disappearance of InP reflections on XRD pattern after ZnS shell formation (Fig. 5c) confirms the presence of an essential strain. The compressive strain results in band gap broadening and PL shift to shorter wavelength. Authors do not take into account do not discuss this phenomenon when addressing the influence of ligand chain length of the sulfur source on ZnS cover growth rate. As the synthesis time is fixed, it means the slower growth rate the thinner ZnS layer and the lower strain impact. When studying influence of the capping ligand length, the ZnS shell becomes even thicker due to the second shell formation and may induce gradual stress relief that results in long wavelength shift in absorption and PL.

Author Response

Point 1: The only remark is that the small-size InP core inside the ZnS shell has a lattice mismatch of 7.7% and should be compressed depending on the shell thickness. Disappearance of InP reflections on XRD pattern after ZnS shell formation (Fig. 5c) confirms the presence of an essential strain. The compressive strain results in band gap broadening and PL shift to shorter wavelength. Authors do not take into account do not discuss this phenomenon when addressing the influence of ligand chain length of the sulfur source on ZnS cover growth rate. As the synthesis time is fixed, it means the slower growth rate the thinner ZnS layer and the lower strain impact. When studying influence of the capping ligand length, the ZnS shell becomes even thicker due to the second shell formation and may induce gradual stress relief that results in long wavelength shift in absorption and PL.
 Response 1: Thank you for giving us your thoughts on things we didn't consider. As the shell thickness increases, there is also an effect due to the compressive deformation mentioned by the reviewer, but in general, the electromagnetic function is delocalized, resulting in a red shift. Shell. Therefore, there seems to be a point in which it is difficult to conclusively explain the effects of both sides. In terms of the lattice deformation mentioned by the reviewer, the following experiment was further conducted. We compared the PL spectra of the QDs with the thickness of the shell and the presence of the GaP inner layer to determine the effect of the lattice strain on the wavelength. The QDs with a GaP inner shell has little change in PL wavelength between single shell and multiple shell, while the spectrum of QD without GaP is not only significant broadening, but also red-shifted by 28nm after coating 2nd shell. We believe that this large difference in optical properties is mainly due to the large lattice constant mismatch between the InP core and the ZnS shell, and the GaP inner layer helped epitaxial ZnS shell growth by mitigating the lattice strain. Therefore, the optical effect due to lattice deformation is considered to be minimized by the GaP inner layer, as the thickness of the ZnS shell increases. ※ Please refer to the attached file for the figure.

Reviewer 2 Report

Please see my uploaded review report.

Author Response

Point 1: How does the QD size distribution look like when different capping ligand is utilized? Will the choice of capping ligand length affect the dispersity of QDs?

Response 1: We were not able to analyze TEM to confirm the QDs size distribution. However, the size distribution of QD can be represented by the full width half maximum (FWHM) values of the PL spectrum. The short chain capping ligand (LA) is observed to possess relatively wide FWHM and low PLQY due to incomplete surface passivation of the QDs. There was no significant difference in the value of FWHM between the remaining ligands (i.e. MA, PA, SA).

 We also fully agree with the reviewer's second question. As can be seen from previous studies (Nano Lett. 2016, 16, 2133−2138, DOI: 10.1021/acs.nanolett.6b00730), long and bulky ligands are more advantageous for dispersing large inorganic particles in organic solvents. Therefore, we think that SA with a long alkyl chain is advantageous in terms of the dispersity of QDs.

Point 2: In lines 173-183, the authors try to explain why their findings contradicted with a previous study. They believe that the length of capping ligand affects the formation of In-P-Zn complex. Experiments using OA as the capping ligand were performed. From Figure 4, it seems that QDs with OA exhibit red-shifted absorption peaks compared to QDs with SA. What is the reason for this absorption peak shift? Also, this experiment does not directly prove that capping ligand affects the formation of In-P-Zn. Could the authors elaborate more on this explanation?

Response 2: QDs with OA showed red-shifted absorption peaks compared to QDs with SA, and the difference in absorption peak between samples was much greater at 140 ℃ and 180 ℃, as the reviewers pointed out. The reaction solution was collected as the temperature of the reactor increased rapidly, and this difference seems to be caused by an error in the sampling temperature. Except for the samples at 140 ℃ and 180 ℃, the absorption peak between samples shows similar values.

 We also agree with the reviewer's second comment. We were not able to conduct a direct analysis of the formation of In-P-Zn complex within the due date of this revision. However, it is well known that the stable In−P−Zn intermediate complexes are also formed at mild temperatures when Zn precursors are introduced, resulting in a retarded nucleation of the InP core [Chem. Mater. 2017, 29, 6346−6355, DOI: 10.1021/acs.chemmater.7b01648]. Therefore, we believe that the In-P-Zn complex was formed in the same way in this synthetic method. Unlike other ligands, the primary excitation absorption peak is exceptionally observed at 100 °C for SA. Based on this, we believe that the nuclear reaction proceeded rapidly because the In-P-Zn complex was not formed due to SA ligand steric hindrance. In terms of steric hindrance, we tried to confirm the reproducibility of the reaction kinetics using an OA ligand having the same alkyl chain length as SA in this experiment. We hope you can understand our situation.

Point 3: In section 3.4, the authors fabricated QLEDs based on the QDs they synthesized. But I did not see a strong correlation of this part to the rest of the manuscript. If one follows the story of this paper, the natural question will be: How does the alkyl chain length influence the performance of final QLED devices? Here the authors only showed one device and it will be hard for readers to appreciate the work. If not compared internally, then another question will be: Is using the QDs fabricated in this work better than previous works?

Response 3: Thanks for the comment on the story line of the paper. We further studied the stability of QDs and the performance of QLEDs along the length of the ligand and will be covered in the next paper. It was confirmed that the shorter the length of the ligand, the higher the current density of the QLED, as shown in below figure. However, the overall device performance was almost similar.

 In this paper, we focused on the synthesis of blue InP QD and fabricated QLED to confirm the possibility for display application. The QDs what we used have low quantum yield (45 %), so the efficiency of the device is low compared to the latest research results. We hope to further improve the quantum efficiency of blue InP QD to expand our research and publish in the near future.

Point 4: QDs/QLEDs for display and lighting industry are hot topics in recent years. It will be nicer to include more up-to-date references in Introduction so that the readers would appreciate better the recent development, for example:

- Ultra-bright solution-processed QLEDs: J. Phys. Chem. Lett. 2019, 10, 2196-2201.

- QDs for display and lighting: Crystals 2019, 9, 59; IEEE J. Sel. Top. Quantum Electron. 2017, 23, 1900611; Nanomaterials 2019, 9, 176; Light Sci. Appl. 2020, 9, 105.

Response 4: Thank you for recommending the latest paper on QD's display application. We also believe that it will be of great help to readers to understand the development trend of the display field of QD. We have additionally cited the following 4 papers in the introduction. (ref. 5 – 8)

  1. He, Z.; Zhang, C.; Dong, Y.; Wu, S.-T. Emerging Perovskite Nanocrystals-Enhanced Solid-State Lighting and Liquid-Crystal Displays. Crystals 2019, 9, 59.
  2. He, Z.; Zhang, C.; Chen, H.; Dong, Y.; Wu, S.-T. Perovskite Downconverters for Efficient, Excellent Color-Rendering, and Circadian Solid-State Lighting. Nanomaterials 2019, 9, 176.
  3. Chen, H.; He, J.; Wu, S.-T. Recent Advances on Quantum-Dot-Enhanced Liquid-Crystal Displays. IEEE J. Sel. Top. Quantum Electron. 2017, 23, 1900611.
  4. Chen, H.; He, Z.; Zhang, D.; Zhang, C.; Ding, Y; Tetard, L.; Wu, S.-T.; Dong, Y. Bright Quantum Dot Light-Emitting Diodes Enabled by Imprinted Speckle Image Holography Nanostructures. J. Phys. Chem. Lett. 2019, 10, 2196−2201.

※ Please see the attachment
